# FROM NOISY TRACES TO STABLE GRADIENTS: BIAS–VARIANCE OPTIMIZED PREFERENCE OPTIMIZATION FOR ALIGNING LARGE REASONING MODELS

## ABSTRACT

Large reasoning models (LRMs) generate intermediate reasoning traces before producing final answers, yielding strong gains on multi-step and mathematical tasks. Yet aligning LRMs with human preferences, a crucial prerequisite for model deployment, remains underexplored. The statistically correct objective for preference alignment requires marginalizing over reasoning traces, but this computation is intractable in practice. A common workaround optimizes a single sampled trajectory, which introduces substantial gradient variance from stochastic trace sampling. To address this challenge, we frame preference optimization for LRMs through the lens of the bias–variance trade-off and propose Bias–Variance Optimized Preference Optimization (BVPO), a simple, drop-in method that mixes two gradient estimators: a high-variance trace-based estimator and a low-variance empty-trace estimator obtained by disabling reasoning trace generation. Our theory shows that BVPO strictly reduces trace-induced variance for any nontrivial mixture, provides a closed-form choice of the mixing weight that minimizes mean-squared error relative to the true marginal gradient, and under standard smoothness and step-size conditions, tightens classical convergence bounds for stochastic gradient descent. Empirically, BVPO improves alignment over the best baseline by up to 7.8 points on AlpacaEval 2 and 6.8 points on Arena-Hard. Despite being trained only on general conversational data, BVPO also boosts reasoning performance for base models by up to 4.0 points on the average of six math reasoning benchmarks. These results identify variance from trace sampling as a key bottleneck and demonstrate that directly optimizing the bias–variance trade-off yields more stable training and stronger overall performance.

## 1 INTRODUCTION

Large reasoning models (LRMs), such as DeepSeek R1, Gemini 2.5, and GPT-o1, scale test-time compute by generating intermediate reasoning traces before producing a final answer (Snell et al., 2025; DeepSeek-AI et al., 2025; Comanici et al., 2025; OpenAI et al., 2024). This explicit deliberation drives large gains on multi-step and mathematically intensive tasks, and reinforcement learning with verifiable rewards further improves such capability (Shao et al., 2024; DeepSeek-AI et al., 2025; Ahmadian et al., 2024; Zeng et al., 2025). While alignment with human preference is a prerequisite for deployment, the alignment of LRMs remains largely unexplored. To the best of our knowledge, there is no systematic treatment of aligning LRMs with human preferences; public discussion is sparse and limited to brief remarks in technical reports accompanying foundational LRMs (DeepSeek-AI et al., 2025; OpenAI et al., 2024). Existing alignment pipelines—from RLHF (Ouyang et al., 2022; Ziegler et al., 2020; Schulman et al., 2017) to direct preference optimization (DPO) (Rafailov et al., 2023b) and its variants (Park et al., 2024; Meng et al., 2024; Ethayarajh et al., 2024; Zhu et al., 2025)—were developed for conventional LLMs that do not externalize lengthy reasoning traces. When applied naively to LRMs, these methods inherit a unique source of instability: *trace-induced gradient variance*.

To explain this, we study preference optimization for LRMs under the trace–answer factorization $\pi_\theta(r, y \mid x) = \pi_\theta(r \mid x)\,\pi_\theta(y \mid x, r)$, where the model first generates a reasoning trace $r$ and then produces the final answer $y$. The statistically correct preference optimization objective compares

*marginal* answer probabilities $\pi_\theta(y \mid x) = \sum_r \pi_\theta(r, y \mid x)$, so that all possible traces leading to the same answer are included. However, this sum spans an exponentially large set of traces, making it computationally infeasible. In practice, it is typically replaced with a single sampled trace, yielding a trace-based preference loss and its gradient, the *trace-based gradient* $g_t$ (DeepSeek-AI et al., 2025). This estimator is easy to compute but highly noisy: long and variable traces produce large fluctuations in joint log-probabilities, which hinder stable optimization.

We propose **Bias–Variance Optimized Preference Optimization (BVPO)** to address the high variance inherent in this trace-based training. BVPO augments the standard trace-based gradient $g_t$ with an empty-trace gradient $g_e$, computed by conditioning the policy on an empty trace. $g_e$ is deterministic with respect to trace sampling and hence has low variance relative to the ideal marginal gradient. BVPO then forms a convex combination, $g_c(\alpha) = \alpha g_t + (1 - \alpha)g_e$, designed to be optimal with respect to the Mean Squared Error (MSE) with respect to the ideal marginal gradient $g_m$. Crucially, MSE can be decomposed into squared bias and variance, providing a principled metric for strategically balancing the high-variance $g_t$ with the low-variance $g_e$. Our analysis guarantees this combined estimator $g_c$ has a lower variance and a strictly better MSE than either component alone for any nontrivial mixture. The resulting MSE reduction directly tightens the SGD convergence bound, providing a principled link between statistical optimality and improved training stability.

Extensive experiments on AlpacaEval 2 (Li et al., 2023) and Arena-Hard (Li et al., 2025) show that BVPO consistently outperforms the best baseline, with gains of up to 7.8 points on AlpacaEval 2 and 6.8 points on Arena-Hard. Because alignment with human preference is typically the final stage before deployment, we also examine whether LRMs' reasoning ability is preserved after alignment. Despite being trained exclusively on general conversational data, BVPO does not degrade, and in fact improves reasoning, raising the base model's average performance across six math reasoning benchmarks by up to 4.0 points, including AIME24/25 (Li et al., 2024), AMC (Li et al., 2024), OlympiadBench (He et al., 2024), Minerva (Lewkowycz et al., 2022), and MATH-500 (Hendrycks et al., 2021). These results indicate that BVPO not only stabilizes the alignment process but also enhances reasoning capabilities. Our key contributions are summarized as follows:

- We identify high gradient variance in aligning LRMs due to stochastic reasoning trace sampling, and propose BVPO, which linearly combines trace-based and low-variance empty-trace gradient estimators, explicitly optimizing the bias–variance trade-off via MSE.

- We prove that BVPO's combined gradient estimator reduces conditional variance induced by trace sampling, derive an MSE-optimal mixing coefficient with domination guarantees, and connect these results to tighter SGD convergence bounds.

- Extensive experiments demonstrate that BVPO achieves gains over the best baseline by up to 6.8 points on Arena-Hard and 7.8 points on AlpacaEval 2. Although trained exclusively on general conversational data, BVPO nevertheless substantially improves the average performance of the base model on six math reasoning benchmarks by up to 4.0 points.

## 2 RELATED WORK

**Large Reasoning Models.** Large reasoning models (LRMs) such as DeepSeek R1 (DeepSeek-AI et al., 2025), Gemini 2.5 (Comanici et al., 2025), and GPT-o1 (OpenAI et al., 2024) mark a new frontier in LLM development. Unlike conventional LLMs, LRMs leverage *test-time scaling* (Snell et al., 2025), generating explicit reasoning traces before producing final answers. This mechanism substantially improves performance on complex, multi-step problems (DeepSeek-AI et al., 2025; Shao et al., 2024). Recent efforts further enhance LRMs' reasoning ability through reinforcement learning with verifiable rewards, especially on mathematically intensive tasks (Shao et al., 2024; DeepSeek-AI et al., 2025; Ahmadian et al., 2024; Zeng et al., 2025). In contrast, to the best of our knowledge, there is no systematic study of aligning LRMs with human preferences, a prerequisite for real-world deployment. Existing discussions are sparse and confined to brief subsections in technical reports of foundation LRMs (DeepSeek-AI et al., 2025). Our work fills this gap by systematically analyzing the alignment challenges unique to LRMs—most notably the high variance induced by long, stochastic reasoning traces, and introducing a principled algorithm to address them.

**Reinforcement Learning from Human Feedback.** Reinforcement Learning from Human Feedback (RLHF) is a foundational approach for aligning large language models (LLMs) with human

preferences (Ouyang et al., 2022; Ziegler et al., 2020; Schulman et al., 2017). Recent efforts focus on bypassing explicit reward model training. A prominent approach is Direct Preference Optimization (DPO) (Rafailov et al., 2023b), which formulates an explicit loss corresponding to the PPO-induced reward. This enables direct fine-tuning without training a reward model. DPO has demonstrated stability and efficiency across diverse applications (Ivison et al., 2024; Tian et al., 2024; Miao et al., 2024). Several extensions refine this framework further: R-DPO (Park et al., 2024) mitigates sensitivity to sequence length, SimPO (Meng et al., 2024) better aligns the objective with the sampling distribution and eliminates the reference model, KTO (Ethayarajh et al., 2024) generalizes preference optimization beyond pairwise comparisons, and TGDPO (Zhu et al., 2025) incorporates token-level reward guidance. However, these methods are developed for conventional LLMs that directly produce final answers. When naively applied to LRMs, which externalize lengthy reasoning traces that reflect the model's internal deliberation and trial-and-error, they face a unique challenge: high gradient variance originating from stochastic trace sampling and large fluctuations in joint log-probabilities. To address this, we propose BVPO, a principled preference optimization method that optimizes bias–variance trade-off, yielding significantly stronger alignment while preserving and even enhancing reasoning performance in math reasoning tasks.

## 3 PREFERENCE OPTIMIZATION FOR LRMS

This section formalizes the problem of aligning Large Reasoning Models with human preferences using preference optimization. We first review the standard DPO objective, highlighting its limitations when applied to LRMs, and then introduce our proposed method.

### 3.1 PRELIMINARIES

**Large Reasoning Models.** An LRM is modeled as a policy $\pi_\theta$ parameterized by $\theta$. Given a prompt $x$, the model first generates an intermediate reasoning trace $r$ and then produces a final answer $y$. This sequential process defines a probability distribution over the complete trajectory $(r, y)$, which factorizes as: $\pi_\theta(r, y \mid x) = \pi_\theta(r \mid x)\, \pi_\theta(y \mid x, r)$. The marginal probability of the final answer $y$ is obtained by summing over all possible reasoning traces: $\pi_\theta(y \mid x) = \sum_r \pi_\theta(r, y \mid x)$.

**Direct Preference Optimization.** DPO (Rafailov et al., 2023b) aligns language models with human preferences by bypassing the explicit reward-modeling stage of traditional RLHF. The key insight is to analytically derive a loss from the Bradley-Terry preference model (Bradley & Terry, 1952), which defines the probability that a response $y^+$ is preferred over $y^-$ as:

$$p(y^+ \succ y^- \mid x) = \sigma\left(r(x, y^+) - r(x, y^-)\right),$$

where $\sigma(\cdot)$ is the sigmoid function and $r(x, y)$ is a latent reward function. DPO defines this reward in terms of the model policy $\pi_\theta$ and a fixed reference policy $\pi_{\text{ref}}$:

$$r(x, y) = \beta \log \frac{\pi_\theta(y \mid x)}{\pi_{\text{ref}}(y \mid x)}.$$

Here, $\beta$ is a temperature parameter that scales the reward difference. Substituting this reward definition into the preference model and maximizing the log-likelihood for a dataset $\mathcal{D}$ of preference tuples $(x, y^+, y^-)$ yields the DPO loss:

$$\mathcal{L}_{\text{DPO}}(\pi_\theta) = -\mathbb{E}_{(x, y^+, y^-) \sim \mathcal{D}} \left[ \log \sigma \left( \beta \log \frac{\pi_\theta(y^+ \mid x)}{\pi_{\text{ref}}(y^+ \mid x)} - \beta \log \frac{\pi_\theta(y^- \mid x)}{\pi_{\text{ref}}(y^- \mid x)} \right) \right].$$

### 3.2 DPO FOR LRMS: IDEAL VS. PRACTICAL OBJECTIVES

Applying the standard DPO framework to LRMs requires adapting its objective to account for reasoning traces. This section formalizes this challenge by contrasting the theoretically ideal objective with its standard, practical approximation.

**Ideal Objective: The Marginal Preference Loss $\mathcal{L}_m$.** The ideal objective for aligning an LRM applies the DPO loss to the marginal probabilities of the final answers. This **marginal preference**

**loss** compares the log-probability ratios of the preferred output $y^+$ and dispreferred output $y^-$ relative to a reference policy $\pi_{\text{ref}}$:

$$\mathcal{L}_m(\theta) = -\mathbb{E}_{(x,y^+,y^-)\sim\mathcal{D}} \left[ \log \sigma \left( \beta \log \frac{\pi_\theta(y^+ \mid x)}{\pi_{\text{ref}}(y^+ \mid x)} - \beta \log \frac{\pi_\theta(y^- \mid x)}{\pi_{\text{ref}}(y^- \mid x)} \right) \right],$$

where the marginal probability is $\pi_\theta(y \mid x) = \sum_r \pi_\theta(r, y \mid x)$. This loss is statistically optimal as it directly models the true preference over final answers. However, computing the marginal probability requires summing over an exponentially large space of possible reasoning traces, rendering this loss computationally intractable.

**Practical Proxy: The Trace-Based Loss $\mathcal{L}_t$.** The standard approach to create a tractable approximation of marginal probabilities for LRMs is to use a single-sample Monte Carlo estimate based on sampled trajectories (DeepSeek-AI et al., 2025). In the case of $\mathcal{L}_m(\theta)$, this yields the **trace-based DPO loss**, which compares the joint probabilities of trace–answer pairs $(r^+, y^+)$ and $(r^-, y^-)$:

$$\mathcal{L}_t(\theta) = \mathbb{E}_{(x,y^\pm,r^\pm)\sim\mathcal{D}_t} \left[ \ell_t(\theta; x, y^\pm, r^\pm) \right], \tag{1}$$

where $\ell_t$ represents the loss associated with a single pair of samples:

$$\ell_t(\theta; x, y^\pm, r^\pm) = -\log \sigma \left( \beta \log \frac{\pi_\theta(r^+, y^+ \mid x)}{\pi_{\text{ref}}(r^+, y^+ \mid x)} - \beta \log \frac{\pi_\theta(r^-, y^- \mid x)}{\pi_{\text{ref}}(r^-, y^- \mid x)} \right).$$

The trace-based gradient, $g_t = \nabla_\theta \ell_t(\theta; x, r^\pm, y^\pm)$, provides a direct optimization signal by operating on full trajectories. While conceptually straightforward, its practical application is challenged by the significant variance of the gradient estimator, which can hinder stable training. This variance is a direct consequence of sampling the reasoning traces $r$. These traces are often long, vary widely in length, and are drawn from a vast search space, causing the joint log-probabilities $\log \pi_\theta(r, y \mid x)$ to fluctuate dramatically across samples and yield a noisy gradient. We further provide empirical evidence in Appendix B that the variance of the log-probabilities and response length with trace generation is much higher than disabling trace generation. This provides concrete evidence that the instability of the trace-based gradient is a significant bottleneck in practice.

### 3.3 BIAS–VARIANCE OPTIMIZED PREFERENCE OPTIMIZATION

The standard trace-based loss $\mathcal{L}_t$ poses a significant challenge to stable alignment because of the high variance of the gradient estimator. To address this issue, we propose **Bias–Variance Optimized Preference Optimization (BVPO)**, an algorithm that creates a more stable training objective by directly managing the bias–variance trade-off. BVPO achieves this by combining the signal from the high-variance $\mathcal{L}_t$ with a novel, low-variance component.

**Empty-Trace Loss $\mathcal{L}_e$.** To directly combat the source of the variance, we introduce the **empty-trace loss**, $\mathcal{L}_e$. This objective bypasses the stochasticity of trace sampling by conditioning the policy on a fixed, empty trace $r = \emptyset$ and applying the DPO objective directly to the final answers. The full loss is the expectation of single-sample losses, $\ell_e$, over the dataset $\mathcal{D}_e$:

$$\mathcal{L}_e(\theta) = \mathbb{E}_{(x,y'^\pm)\sim\mathcal{D}_e} \left[ \ell_e(\theta; x, y'^\pm) \right],$$

where $\ell_e$ is defined for a single preference pair as:

$$\ell_e(\theta; x, y'^\pm) = -\log \sigma \left( \beta \log \frac{\pi_\theta(r = \emptyset, y'^+ \mid x)}{\pi_{\text{ref}}(r = \emptyset, y'^+ \mid x)} - \beta \log \frac{\pi_\theta(r = \emptyset, y'^- \mid x)}{\pi_{\text{ref}}(r = \emptyset, y'^- \mid x)} \right).$$

The gradient of this single-sample loss, $g_e = \nabla_\theta \ell_e$, exhibits lower variance because it avoids sampling from the vast space of reasoning traces. The trade-off is a potentially higher bias, as it ignores the reasoning process.

**Combined BVPO Loss $\mathcal{L}_c$.** To exploit the accuracy of the trace-based estimator while mitigating its variance with the stability of the empty-trace estimator, we define the **combined BVPO loss** as their convex combination:

$$\mathcal{L}_c(\theta) = \alpha \mathcal{L}_t(\theta) + (1 - \alpha) \mathcal{L}_e(\theta), \tag{2}$$

where $\alpha \in [0, 1]$ is a hyperparameter controlling the interpolation. The resulting gradient estimator $g_c$ is a weighted average of the individual estimators:

$$g_c = \alpha g_t + (1 - \alpha) g_e.$$

This formulation provides principled control over the bias–variance trade-off. By tuning $\alpha$, one can obtain a combined estimator $g_c$ that improves upon both $g_t$ and $g_e$. In Section 4, we formally prove its variance-reduction property and show that $g_c$ achieves a more favorable bias–variance balance than either component alone.

**Practical Implementation of BVPO.** Given a prompt dataset $\mathcal{D} = \{x_i\}_{i=1}^N$, we construct the preference dataset for the **Trace-Based Loss** by sampling from $\pi_{\text{ref}}$, yielding $\mathcal{D}_t = \{(x_i, r_i^\pm, y_i^\pm)\}_{i=1}^N$. For the **Empty-Trace Loss**, we disable reasoning trace generation by appending "`<think></think>`" to each input prompt $x_i$, producing the preference dataset $\mathcal{D}_e = \{(x_i, y_i'^\pm)\}_{i=1}^N$. Preference comparisons are made solely on the final responses $y$, since reasoning traces are often long, noisy, and include trial-and-error steps. This mirrors prior practice in DeepSeek-AI et al. (2025), where PPO was applied with rewards based only on $y$. Our mixed-gradient estimator $g_c$ is agnostic to the preference optimization algorithm. In practice, we instantiate it with the widely used DPO objective, yielding the combined BVPO loss in Equation (2).

## 4 THEORETICAL ANALYSIS

We now ask: does the mixed estimator $g_c$ provably improve over its components $g_t$ and $g_e$? We show that it achieves variance reduction w.r.t. trace sampling (Theorem 1), optimal MSE guarantees (Theorem 2), and these statistical gains yield stronger convergence for SGD (Theorems 3 and 4).

### 4.1 REDUCTION OF CONDITIONAL VARIANCE INDUCED BY TRACE SAMPLING

The high variance of the trace-based estimator $g_t$ often impedes stable optimization. To mitigate this, our combined estimator $g_c$ incorporates the low-variance empty-trace estimator $g_e$, reducing variance while retaining the directional information of $g_t$, as shown below.

**Theorem 1** (Conditional Variance Reduction for Trace Sampling). *The trace-based estimator $g_t$ is a random variable dependent on a sampled trace $r^\pm$, while the empty-trace estimator $g_e$ is deterministic with respect to trace sampling. For a vector-valued gradient $g$, its scalar variance is defined as the trace of its covariance matrix, $\text{Var}(g) := \text{tr}(\text{Cov}(g)) = \mathbb{E}[\|g - \mathbb{E}[g]\|_2^2]$.*

*The combined estimator $g_c = \alpha g_t + (1 - \alpha) g_e$, with a fixed mixing coefficient $\alpha \in [0, 1]$, has a conditional variance (with respect to trace sampling) that is bounded above by that of $g_t$. Specifically, for any data sample $(x, y^\pm, y'^\pm)$:*

$$\text{Var}_{r^\pm}(g_c \mid x, y^\pm, y'^\pm) = \alpha^2 \text{Var}_{r^\pm}(g_t \mid x, y^\pm) \leq \text{Var}_{r^\pm}(g_t \mid x, y^\pm).$$

*Consequently, the expected conditional variance is also bounded:*

$$\mathbb{E}_{x,y^\pm,y'^\pm}[\text{Var}_{r^\pm}(g_c \mid x, y^\pm, y'^\pm)] \leq \mathbb{E}_{x,y^\pm}[\text{Var}_{r^\pm}(g_t \mid x, y^\pm)].$$

The proof of Theorem 1 is given in Appendix A.1. In this theorem, the first inequality is strict whenever $\alpha \in (0, 1)$ and $\text{Var}_{r^\pm}(g_t \mid x, y^\pm) > 0$.

Theorem 1 formalizes a key benefit of our approach: incorporating the gradient estimator $g_e$ guarantees to reduce the variance stemming from trace sampling. The degree of this reduction is controlled by $\alpha$. However, this benefit comes with a trade-off. While a smaller $\alpha$ suppresses variance, it may increase the bias with respect to the true marginal gradient by shifting the estimator's mean. This introduces the classic bias-variance trade-off, which we analyze in the next section.

### 4.2 OPTIMAL COMBINATION OF GRADIENT ESTIMATORS BY MSE MINIMIZATION

To determine the best balance between bias and variance, we seek the value of $\alpha$ that minimizes the **mean squared error (MSE)** of $g_c$ with respect to the true marginal gradient, $\mu = \nabla_\theta \mathcal{L}_m(\theta)$. The MSE provides a comprehensive measure of estimator quality, as it simultaneously penalizes both variance and systematic deviation from the target gradient.

**Theorem 2** (Optimal Convex Combination of Gradient Estimators). *For two estimators $g_t$ and $g_e$ of the true marginal gradient $\mu := \nabla_\theta \mathcal{L}_m(\theta)$, assume the estimators have finite first and second moments, with bias vectors $b_t := \mathbb{E}[g_t] - \mu$, $b_e := \mathbb{E}[g_e] - \mu$, and covariance matrices $\Sigma_t = \mathrm{Cov}(g_t)$, $\Sigma_e = \mathrm{Cov}(g_e)$, and*

$$\Sigma_{te} = \mathrm{Cov}(g_t, g_e) = \mathbb{E}[(g_t - \mathbb{E}[g_t])(g_e - \mathbb{E}[g_e])^\top].$$

*The combined estimator $g_c(\alpha) = \alpha g_t + (1-\alpha)g_e$ for $\alpha \in [0,1]$ has an MSE defined by*

$$\mathrm{MSE}(g_c(\alpha)) := \mathbb{E}[\|g_c(\alpha) - \mu\|^2].$$

*If $\mathbb{E}[\|g_t - g_e\|^2] > 0$, then the unconstrained value of $\alpha$ that minimizes this MSE is:*

$$\alpha_{\mathrm{unc}} = \frac{\mathrm{tr}(\Sigma_e - \Sigma_{te}) + \|b_e\|^2 - b_t^\top b_e}{\mathbb{E}[\|g_t - g_e\|^2]},$$

*and the optimal parameter within the valid interval is $\alpha^\star = \max(0, \min(1, \alpha_{\mathrm{unc}}))$. If $\mathbb{E}[\|g_t - g_e\|^2] = 0$, then any $\alpha^\star \in [0,1]$ is optimal.*

*This optimal estimator is guaranteed to be no worse than the better of the two individual estimators:*

$$\mathrm{MSE}(g_c(\alpha^\star)) \leq \min\big\{\mathrm{MSE}(g_t), \mathrm{MSE}(g_e)\big\}.$$

The proof of this theorem is given in Appendix A.2. Theorem 2 provides a principled method for finding the optimal estimator $g_c(\alpha^\star)$ among all possible convex combinations. The guarantee is powerful: our combined estimator can never underperform the best-performing individual estimator in terms of MSE. In fact, the improvement is typically strict, as formalized below.

**Corollary 1** (Strict Improvement Over $g_t$). *Assume $\mathbb{E}[\|g_t - g_e\|^2] > 0$. If the optimal coefficient $\alpha^\star$ lies in the open interval $(0,1)$, then the combined estimator strictly dominates $g_t$:*

$$\mathrm{MSE}(g_c(\alpha^\star)) < \mathrm{MSE}(g_t).$$

*Consequently, unless the optimum lies at $\alpha^\star = 1$ or $g_t \equiv g_e$, $g_c(\alpha^\star)$ yields a strict improvement upon $g_t$ in MSE.*

By symmetry, an analogous result holds when comparing against $g_e$. If $\mathbb{E}[\|g_t - g_e\|^2] > 0$ and the optimal coefficient $\alpha^\star$ lies in $(0,1)$, then $\mathrm{MSE}(g_c(\alpha^\star)) < \mathrm{MSE}(g_e)$. Thus, unless $\alpha^\star = 0$ or $g_t \equiv g_e$, the combined estimator $g_c(\alpha^\star)$ yields a strict improvement upon $g_e$ as well.

This statistical optimality of Theorem 2 and Corollary 1 has direct algorithmic implications. Specifically, the property that the combined estimator $g_c(\alpha^\star)$ minimizes the mean squared error with respect to the true marginal gradient implies that it provides the most accurate gradient estimate on average, balancing variance and bias. In stochastic optimization, the quality of the gradient estimate at each iteration governs both the stability and speed of convergence. An estimator with lower MSE yields update directions that are more faithful to the true gradient, simultaneously reducing stochastic noise and systematic drift. With the MSE-optimal estimator $g_c(\alpha^\star)$, we therefore expect more stable optimization. The following section formalizes this intuition by analyzing the convergence bounds for SGD using our combined estimator.

### 4.3 Convergence Guarantees for SGD

Having established that our estimator is statistically optimal in terms of MSE, we now connect this property to its algorithmic performance. In stochastic gradient descent (SGD), convergence is fundamentally limited by the quality of the gradient estimates. To formalize this, we present the following convergence Theorem 3. The theorem and its proof are adapted from Karimireddy et al. (2022), which builds upon the well-established analysis for SGD with biased gradients (e.g., Ghadimi & Lan, 2013; Ajalloeian & Stich, 2020). This theorem is pivotal: it reveals that the convergence bound is governed by the estimator's squared bias and variance. Since MSE is precisely the sum of these two error terms (see Equation (5)), our approach of minimizing the MSE is explicitly designed to minimize the dominant factors that limit the algorithm's performance.

**Theorem 3** (SGD Convergence under BVPO Estimator). *Let $\mathcal{L}_m : \mathbb{R}^d \to \mathbb{R}$ be an L-smooth function with minimum value of $\mathcal{L}^*$. Consider stochastic gradient descent: $\theta_{k+1} = \theta_k - \eta\, g_c(\theta_k)$, where $g_c(\theta_k)$ is the stochastic combined gradient estimator at iterate $\theta_k$. Let $\mu_k := \nabla_\theta \mathcal{L}_m(\theta_k)$ denote the true marginal gradient. Define the conditional expectation and variance*

$$\mathbb{E}_k[\cdot] := \mathbb{E}[\cdot \mid \theta_k], \qquad \mathrm{Var}_k(g_c) := \mathbb{E}_k\big[\|g_c(\theta_k) - \mathbb{E}_k[g_c(\theta_k)]\|^2\big],$$

*and the conditional bias vector $\mathrm{Bias}_k := \mathbb{E}_k[g_c(\theta_k)] - \mu_k$. If the constant step size satisfies $\eta \le 1/L$, then the averaged squared norm of the true gradient satisfies the exact bound:*

$$\mathbb{E}\left[\frac{1}{K}\sum_{k=0}^{K-1}\|\nabla_\theta\mathcal{L}_m(\theta_k)\|^2\right] \le \frac{2}{K\eta}\big(\mathcal{L}_m(\theta_0) - \mathbb{E}[\mathcal{L}_m(\theta_K)]\big) + \frac{1}{K}\sum_{k=0}^{K-1}\mathbb{E}\big[\|\mathrm{Bias}_k\|^2 + \eta L\mathrm{Var}_k(g_c)\big]. \tag{3}$$

*Furthermore, if there exist uniform bounds*

$$\|\mathrm{Bias}_k\| \le B_c, \qquad \mathrm{Var}_k(g_c) \le \sigma_c^2, \quad \forall k,$$

*then, using $\mathbb{E}[\mathcal{L}_m(\theta_K)] \ge \mathcal{L}^*$, the bound simplifies to*

$$\mathbb{E}\left[\frac{1}{K}\sum_{k=0}^{K-1}\|\nabla_\theta\mathcal{L}_m(\theta_k)\|^2\right] \le \frac{2(\mathcal{L}_m(\theta_0) - \mathcal{L}^*)}{K\eta} + B_c^2 + \eta L\,\sigma_c^2. \tag{4}$$

The proof of this theorem is given in Appendix A.3. Theorem 3 gives a standard convergence guarantee for SGD. In particular, the last two terms of Equation (4) define an *error floor* determined by the squared bias and variance of the gradient estimator $g_c$. This means that, although SGD converges toward the optimum at the usual $\mathcal{O}(1/K)$ rate, its final accuracy is limited by the bias–variance tradeoff of $g_c$.

To reduce this error floor, we consider an adaptive estimator $g_c(\alpha_k, \theta_k) = \alpha_k g_t(\theta_k) + (1 - \alpha_k)g_e(\theta_k)$, where the mixing weight $\alpha_k$ can be tuned at each iteration. Substituting this estimator into the general bound from Equation (3) gives

$$\mathbb{E}\left[\frac{1}{K}\sum_{k=0}^{K-1}\|\nabla_\theta\mathcal{L}_m(\theta_k)\|^2\right] \le \frac{2}{K\eta}\big(\mathcal{L}_m(\theta_0) - \mathbb{E}[\mathcal{L}_m(\theta_K)]\big)$$

$$+ \frac{1}{K}\sum_{k=0}^{K-1}\mathbb{E}\big[\|\mathrm{Bias}_k(\alpha_k)\|^2 + \eta L\,\mathrm{Var}_k(g_c(\alpha_k))\big].$$

The key observation is that the per-step error contribution

$$\|\mathrm{Bias}_k(\alpha_k)\|^2 + \eta L\,\mathrm{Var}_k(g_c(\alpha_k))$$

can itself be minimized. In particular, when $\eta L = 1$, the standard bias–variance decomposition,

$$\mathrm{MSE}_k(g_c(\alpha, \theta_k)) = \mathbb{E}_k\big[\|g_c(\alpha, \theta_k) - \mu_k\|^2\big] = \|\mathrm{Bias}_k(g_c(\alpha))\|^2 + \mathrm{Var}_k(g_c(\alpha)), \tag{5}$$

shows that the optimal choice $\alpha_k^\star$ is exactly the one that minimizes the MSE, as shown in Section 4.2. Details of deriving Equation (5) are given in Appendix A.4.

This establishes a direct link between statistical and algorithmic performance: the error floor in the SGD bound, $B_c^2 + \eta L\,\sigma_c^2$, is essentially the MSE, $B_c^2 + \sigma_c^2$, up to the factor $\eta L$, which reflects the algorithm's sensitivity to gradient noise. When $\eta L \approx 1$, minimizing MSE is therefore equivalent to minimizing the convergence error. The following theorem formalizes this intuition.

**Theorem 4** (Optimality of the MSE-Minimal Estimator for SGD). *Let the conditions of Theorem 3 hold. At each iteration $k$, the per-step error in the convergence bound is $E_k(\alpha) := \|\mathrm{Bias}_k(g_c(\alpha))\|^2 + \eta L\,\mathrm{Var}_k(g_c(\alpha))$. Let $\alpha_k^\star$ be the weight that minimizes the conditional Mean Squared Error $\mathrm{MSE}_k(g_c(\alpha, \theta_k))$.*

*If the learning rate and smoothness constant satisfy $\eta L = 1$, then the MSE-optimal weight $\alpha_k^\star$ also minimizes the per-step convergence error: $E_k(\alpha_k^\star) \le E_k(\alpha)$ for all $\alpha \in [0, 1]$.*

The proof of this theorem is given in Appendix A.5. Theorem 4 makes explicit the link between statistical and algorithmic optimality, and provides a simple but powerful conclusion: under a standard choice of learning rate, **the estimator that is statistically optimal (MSE-minimal) is precisely the one that is algorithmically optimal (minimizing the convergence error at each step).**

Table 1: Experiment results on alignment benchmarks: Arena-Hard (Li et al., 2025) and AlpacaEval 2 (Li et al., 2023). LC Win Rate denotes length-controlled win rate.

| Method | Thinking | | | NoThinking | | |
| --- | --- | --- | --- | --- | --- | --- |
| | Arena-Hard | AlpacaEval 2 | | Arena-Hard | AlpacaEval 2 | |
| | Win Rate(%) | Win Rate(%) | LC Win Rate(%) | Win Rate(%) | Win Rate(%) | LC Win Rate(%) |
| R1-Qwen-7B | 16.3 | 15.7 | 18.4 | 16.7 | 15.2 | 17.0 |
| SimPO | 19.0 | 17.8 | 20.2 | 17.2 | 19.1 | 20.3 |
| DPO | 19.1 | 18.3 | 20.4 | 17.7 | 19.3 | 20.7 |
| BVPO | **24.2** | **26.1** | **25.5** | **24.5** | **25.2** | **25.2** |
| **Method** | **Thinking** | | | **NoThinking** | | |
| | Arena-Hard | AlpacaEval 2 | | Arena-Hard | AlpacaEval 2 | |
| | Win Rate(%) | Win Rate(%) | LC Win Rate(%) | Win Rate(%) | Win Rate(%) | LC Win Rate(%) |
| R1-Qwen-1.5B | 4.4 | 5.4 | 6.3 | 5.5 | 6.9 | 6.9 |
| SimPO | 5.5 | 6.2 | 8.4 | 4.5 | 4.6 | 3.8 |
| DPO | 5.1 | 6.4 | 8.0 | 7.2 | 7.8 | 7.1 |
| BVPO | **8.7** | **8.6** | **9.4** | **8.0** | **10.6** | **10.3** |
| **Method** | **Thinking** | | | **NoThinking** | | |
| | Arena-Hard | AlpacaEval 2 | | Arena-Hard | AlpacaEval 2 | |
| | Win Rate(%) | Win Rate(%) | LC Win Rate(%) | Win Rate(%) | Win Rate(%) | LC Win Rate(%) |
| R1-0528-Qwen3-8B | 65.4 | 48.7 | 39.6 | 65.2 | 37.5 | 31.8 |
| SimPO | 69.2 | 49.1 | 44.9 | 62.1 | 41.2 | 41.5 |
| DPO | 68.7 | 48.9 | 44.3 | 61.6 | 40.3 | 40.0 |
| BVPO | **71.5** | **50.6** | **45.9** | **66.8** | **46.6** | **48.4** |

## 5 EXPERIMENTS

In this section, we empirically validate Bias–Variance Optimized Preference Optimization (BVPO) on three large reasoning models, focusing on whether the combined gradient estimator $g_c$ improves alignment without degrading reasoning ability.

### 5.1 EXPERIMENT SETTINGS

**Models and Training Settings.** We conduct experiments on three LRMs: DeepSeek-R1-Distill-Qwen-7B, DeepSeek-R1-Distill-Qwen-1.5B, and DeepSeek-R1-0528-Qwen3-8B (DeepSeek-AI et al., 2025). These models are trained with chain-of-thought reasoning data using SFT from Qwen 2.5 (Qwen et al., 2025) and Qwen 3 (Yang et al., 2025) base models, and have not been trained by RLHF. We use prompts from the UltraFeedback dataset (Cui et al., 2024) and let each model generate 5 responses with a temperature of 0.8. These responses are then ranked using the ArmoRM model (Wang et al., 2024). The response score is calculated only using the final answer part of the response, following DeepSeek-AI et al. (2025). The highest and lowest-ranked responses are selected as the preferred and dispreferred samples, respectively. We use our $g_c$ with the DPO objective to implement our BVPO and compare its performance against the original base models and two state-of-the-art preference optimization methods: DPO (Rafailov et al., 2023a) and SimPO (Meng et al., 2024).

**Evaluation Benchmarks.** We evaluate alignment performance on two widely used open-ended instruction-following benchmarks: Arena-Hard (Li et al., 2025) and AlpacaEval 2 (Li et al., 2023), which measure response quality across diverse prompts. For Arena-Hard, we report the win rate against GPT-4-0314. For AlpacaEval 2, we report both the win rate and the length-controlled win rate against GPT-4 Turbo. We assess LRMs in two modes: the standard reasoning mode, denoted *Thinking*; and suppressing reasoning trace generation by appending "`<think></think>`" to the input prompt, denoted *NoThinking*, reflecting scenarios in which users prefer instant responses without reasoning. To examine whether reasoning capabilities are preserved after alignment, we further evaluate on six widely used math-reasoning benchmarks: AIME 2024, AIME 2025, AMC (Li et al., 2024), Minerva (Lewkowycz et al., 2022), OlympiadBench (He et al., 2024), and MATH-500 (Hendrycks et al., 2021). We report avg@32 accuracy for AIME 2024, AIME 2025, and AMC

Table 2: Experiment results on math reasoning benchmarks: AIME 2024, AIME 2025, AMC (Li et al., 2024), Minerva (Lewkowycz et al., 2022), OlympiadBench (He et al., 2024), and MATH-500 (Hendrycks et al., 2021).

| Method | AIME 24 | AIME 25 | AMC | MATH-500 | Minerva | Olympiadbench | Avg. |
|---|---|---|---|---|---|---|---|
| R1-Qwen-7B | 56.3 | 40.3 | 79.7 | 89.2 | 40.1 | 57.5 | 60.5 |
| SimPO | 55.8 | 38.2 | 80.7 | 88.2 | 41.2 | 58.4 | 60.4 |
| DPO | 55.0 | 40.7 | 80.8 | 89.8 | 40.8 | 59.0 | 61.0 |
| BVPO | 58.4 | 41.0 | 81.2 | 89.4 | 43.0 | 60.9 | **62.3** |
| **Method** | **AIME 24** | **AIME 25** | **AMC** | **MATH-500** | **Minerva** | **Olympiadbench** | **Avg.** |
| R1-Qwen-1.5B | 28.6 | 21.7 | 62.2 | 81.8 | 29.0 | 44.9 | 44.7 |
| SimPO | 30.2 | 23.3 | 62.5 | 82.6 | 30.5 | 46.2 | 45.9 |
| DPO | 31.7 | 23.4 | 64.9 | 84.0 | 33.8 | 48.7 | 47.8 |
| BVPO | 34.4 | 24.4 | 65.1 | 83.0 | 35.3 | 50.1 | **48.7** |
| **Method** | **AIME 24** | **AIME 25** | **AMC** | **MATH-500** | **Minerva** | **Olympiadbench** | **Avg.** |
| R1-0528-Qwen3-8B | 73.1 | 66.0 | 91.8 | 96.4 | 47.1 | 73.5 | 74.7 |
| SimPO | 73.9 | 66.1 | 91.0 | 96.4 | 47.5 | 76.0 | 75.2 |
| DPO | 73.6 | 65.9 | 91.0 | 97.6 | 47.1 | 76.0 | 75.2 |
| BVPO | 76.3 | 68.0 | 91.7 | 96.8 | 46.7 | 76.9 | **76.1** |

due to their small test set sizes, and pass@1 for the remaining benchmarks. All evaluations use a temperature of 0.6. Additional experimental details are provided in Appendix C.

## 5.2 MAIN RESULTS

**BVPO Consistently Improves Alignment.** Table 1 reports alignment results on AlpacaEval 2 (Li et al., 2023) and Arena-Hard (Li et al., 2025). Across both benchmarks, BVPO consistently surpasses the best baselines. In *Thinking* mode, BVPO improves AlpacaEval 2 win rate by up to 7.8 points and the length-controlled win rate by up to 5.1 points, and increases the Arena-Hard win rate by up to 5.1 points. In *NoThinking* mode, BVPO yields gains of up to 6.8 points on Arena-Hard and up to 5.9 win rate and 6.9 length-controlled win rate points on AlpacaEval 2. These results demonstrate BVPO's effectiveness by leveraging the bias-variance optimal gradient estimator.

**Preference Optimization Preserves and Improves Reasoning Ability.** Because human preference alignment is typically the final tuning stage before deployment, it is crucial that preference optimization not erode LRMs' reasoning ability acquired from earlier reinforcement learning with verifiable rewards. As shown in Table 2, evaluated on six widely adopted math reasoning benchmarks (AIME 2024, AIME 2025, AMC (Li et al., 2024), Minerva (Lewkowycz et al., 2022), OlympiadBench (He et al., 2024), and MATH-500 (Hendrycks et al., 2021)), both DPO and BVPO maintain and improve LRMs' reasoning performance. Notably, BVPO achieves nontrivial gains of up to 4.0 average points across these benchmarks over the base model and, on average, exceeds DPO. These findings indicate that preference alignment using general conversational (non–math-specialized) training data does not sacrifice, and can in fact strengthen reasoning ability for LRMs.

## 6 CONCLUSION

We have studied preference optimization for LRMs, where the statistically correct marginal objective is intractable and practical single-trace surrogates suffer from high-variance gradients. We propose **BVPO**, which combines the standard trace-based gradient $g_t$ with a low-variance empty-trace gradient $g_e$ via convex combination: $g_c = \alpha g_t + (1 - \alpha)g_e$. Theoretically, we prove that $g_c$ reduces variance induced from trace sampling, and that with the optimal $\alpha$, its MSE never exceeds that of $g_t$, yielding sharper SGD convergence under standard assumptions. Empirically, BVPO consistently improves alignment over DPO on AlpacaEval 2 and Arena-Hard, while also enhancing reasoning performance on math reasoning benchmarks. These results highlight trace sampling variance as a key bottleneck for LRM alignment and show that explicitly optimizing the bias–variance trade-off yields both stability and quality improvements.

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

# A    PROOFS OF THEORETICAL RESULTS

## A.1    PROOF OF THEOREM 1

*Proof.* We consider the conditional variance for fixed data sample $(x, y^{\pm})$ and $(x, y'^{\pm})$. The estimator $g_e$ is deterministic with respect to the trace sampling distribution $p(r^{\pm} \mid x, y'^{\pm})$. As the variance of a deterministic quantity is zero, $\mathrm{Var}_{r^{\pm}}(g_e \mid x, y'^{\pm}) = 0$.

The conditional variance of the combined estimator $g_c$ is derived as follows, using the property that for a random vector $X$ and constant vector $b$, $\mathrm{Var}(aX + b) = a^2\mathrm{Var}(X)$:

$$
\begin{aligned}
\mathrm{Var}_{r^{\pm}}\big(g_c \mid x, y^{\pm}, y'^{\pm}\big) &= \mathrm{Var}_{r^{\pm}}\big(\alpha g_t + (1-\alpha)g_e \,\big|\, x, y^{\pm}, y'^{\pm}\big) \\
&= \mathrm{Var}_{r^{\pm}}\big(\alpha g_t \mid x, y^{\pm}\big) + \mathrm{Var}_{r^{\pm}}\big((1-\alpha)g_e \mid x, y'^{\pm}\big) \\
&\quad + 2\,\mathrm{tr}(\mathrm{Cov}_{r^{\pm}}\big(\alpha g_t, (1-\alpha)g_e \,\big|\, x, y^{\pm}, y'^{\pm}\big)) \\
&= \alpha^2 \,\mathrm{Var}_{r^{\pm}}\big(g_t \mid x, y^{\pm}\big) \\
&\quad + (1-\alpha)^2 \underbrace{\mathrm{Var}_{r^{\pm}}\big(g_e \mid x, y'^{\pm}\big)}_{=\,0} \\
&\quad + 2\alpha(1-\alpha)\underbrace{\mathrm{tr}(\mathrm{Cov}_{r^{\pm}}\big(g_t, g_e \mid x, y^{\pm}, y'^{\pm}\big))}_{=\,0}
\end{aligned}
$$

$$= \alpha^2 \operatorname{Var}_{r^\pm}(g_t \mid x, y^\pm).$$

In the third equality of the above equation, the variance and covariance with respect to $g_e$ is 0 due to $g_e$ is independent of $r^\pm$ or constant under $r^\pm$. Since $\alpha \in [0, 1]$, it follows that $\alpha^2 \leq 1$. Because variance is non-negative, we have:

$$\alpha^2 \operatorname{Var}_{r^\pm}(g_t \mid x, y^\pm) \leq \operatorname{Var}_{r^\pm}(g_t \mid x, y^\pm).$$

This directly implies the first inequality of this theorem:

$$\operatorname{Var}_{r^\pm}(g_c \mid x, y^\pm, y'^\pm) \leq \operatorname{Var}_{r^\pm}(g_t \mid x, y^\pm).$$

Taking the expectation of this inequality with respect to the data distribution $p(x, y^\pm)$ yields the second inequality of this theorem, which completes the proof. $\qquad\square$

### A.2 PROOF OF THEOREM 2

*Proof.* The proof strategy is to express the MSE as a convex quadratic function of $\alpha$ and find its minimum. The bias and scalar variance of the combined estimator $g_c(\alpha)$ are used to define the MSE. The bias vector is:

$$\begin{aligned}
\operatorname{Bias}\big(g_c(\alpha)\big) &:= \mathbb{E}\big[g_c(\alpha)\big] - \mu, \\
&= \mathbb{E}\big[\alpha g_t + (1 - \alpha)g_e\big] - \mu \\
&= \alpha \mathbb{E}[g_t] + (1 - \alpha)\mathbb{E}[g_e] - \mu \\
&= \alpha\big(\mathbb{E}[g_t] - \mu\big) + (1 - \alpha)\big(\mathbb{E}[g_e] - \mu\big) \\
&= \alpha\, b_t + (1 - \alpha)\, b_e,
\end{aligned} \tag{6}$$

where $\mu := \nabla_\theta \mathcal{L}_m(\theta)$. The scalar variance is:

$$\begin{aligned}
\operatorname{Var}(g_c(\alpha)) &:= \mathbb{E}\big[\|g_c(\alpha) - \mathbb{E}[g_c(\alpha)]\|^2\big] \\
&= \mathbb{E}\big[\|\alpha(g_t - \mathbb{E}[g_t]) + (1 - \alpha)(g_e - \mathbb{E}[g_e])\|^2\big] \\
&= \alpha^2 \mathbb{E}[\|g_t - \mathbb{E}[g_t]\|^2] + (1 - \alpha)^2 \mathbb{E}[\|g_e - \mathbb{E}[g_e]\|^2] + 2\alpha(1 - \alpha)\operatorname{tr}(\operatorname{Cov}(g_t, g_e)) \\
&= \alpha^2 \operatorname{Var}(g_t) + (1 - \alpha)^2 \operatorname{Var}(g_e) + 2\alpha(1 - \alpha)\operatorname{tr}(\Sigma_{te}) \\
&= \alpha^2 \operatorname{tr}(\Sigma_t) + (1 - \alpha)^2 \operatorname{tr}(\Sigma_e) + 2\alpha(1 - \alpha)\operatorname{tr}(\Sigma_{te}).
\end{aligned} \tag{7}$$

The MSE is the squared norm of the bias plus the trace of the variance:

$$\begin{aligned}
\operatorname{MSE}(g_c(\alpha)) &= \mathbb{E}\big[\|g_c(\alpha) - \mu\|^2\big] \\
&= \mathbb{E}\Big[\big\|\underbrace{g_c(\alpha) - \mathbb{E}[g_c(\alpha)]}_{:=\Delta} + \underbrace{\mathbb{E}[g_c(\alpha)] - \mu}_{:=b}\big\|^2\Big] \qquad \text{(add \& subtract } \mathbb{E}[g_c(\alpha)]) \\
&= \mathbb{E}\big[\|\Delta\|^2\big] + 2\,\mathbb{E}\big[\Delta^\top b\big] + \|b\|^2 \qquad \text{(expand } \|x + y\|^2) \\
&= \mathbb{E}\big[\|\Delta\|^2\big] + \|b\|^2 \qquad (\mathbb{E}[\Delta] = 0 \Rightarrow \mathbb{E}[\Delta^\top b] = b^\top \mathbb{E}[\Delta] = 0) \\
&= \|\mathbb{E}[g_c(\alpha)] - \mu\|^2 + \mathbb{E}\big[\|\Delta\|^2\big] \qquad (b = \mathbb{E}[g_c(\alpha)] - \mu) \\
&= \|\operatorname{Bias}(g_c(\alpha))\|^2 + \operatorname{Var}(g_c(\alpha)) \qquad (\operatorname{Var}(g_c(\alpha)) = \mathbb{E}[\|\Delta\|^2]).
\end{aligned}$$

Then by Equations (6) and (7),

$$\begin{aligned}
\operatorname{MSE}(g_c(\alpha)) &= \alpha^2 \|b_t\|^2 + (1 - \alpha)^2 \|b_e\|^2 + 2\alpha(1 - \alpha)b_t^\top b_e \\
&\quad + \alpha^2 \operatorname{tr}(\Sigma_t) + (1 - \alpha)^2 \operatorname{tr}(\Sigma_e) + 2\alpha(1 - \alpha)\operatorname{tr}(\Sigma_{te}) \\
&= \alpha^2 \|b_t\|^2 + (1 - 2\alpha + \alpha^2)\|b_e\|^2 + 2\alpha b_t^\top b_e - 2\alpha^2 b_t^\top b_e \\
&\quad + \alpha^2 \operatorname{tr}(\Sigma_t) + (1 - 2\alpha + \alpha^2)\operatorname{tr}(\Sigma_e) + 2\alpha \operatorname{tr}(\Sigma_{te}) - 2\alpha^2 \operatorname{tr}(\Sigma_{te}) \\
&= \alpha^2 \big[\|b_t\|^2 + \|b_e\|^2 - 2b_t^\top b_e + \operatorname{tr}(\Sigma_t) + \operatorname{tr}(\Sigma_e) - 2\operatorname{tr}(\Sigma_{te})\big] \\
&\quad + \alpha \big[-2\|b_e\|^2 + 2b_t^\top b_e - 2\operatorname{tr}(\Sigma_e) + 2\operatorname{tr}(\Sigma_{te})\big]
\end{aligned}$$

$$+ \|b_e\|^2 + \mathrm{tr}(\Sigma_e).$$

This expression is a quadratic function of $\alpha$, which can be written as

$$\mathrm{MSE}(g_c(\alpha)) = A\alpha^2 - 2B\alpha + C.$$

By collecting the coefficients for the powers of $\alpha$, we find:

$$A = \|b_t - b_e\|^2 + \mathrm{tr}(\Sigma_t + \Sigma_e - 2\Sigma_{te}) = \|b_t - b_e\|^2 + \mathrm{Var}(g_t - g_e) = \mathbb{E}[\|g_t - g_e\|^2],$$

$$B = \|b_e\|^2 - b_t^\top b_e + \mathrm{tr}(\Sigma_e - \Sigma_{te}).$$

Since $A \geq 0$, the MSE is therefore a convex parabola in $\alpha$. If $A > 0$, the unconstrained minimizer is found by setting the derivative $d(\mathrm{MSE})/d\alpha = 2A\alpha - 2B$ to zero, which yields:

$$\alpha_{\mathrm{unc}} = \frac{B}{A}.$$

This matches the expression in the theorem. To ensure the solution lies in the valid interval, the optimal constrained parameter is $\alpha^\star = \max(0, \min(1, \alpha_{\mathrm{unc}}))$. If $A = 0$, then $\mathbb{E}[\|g_t - g_e\|^2] = 0$, implying $g_t = g_e$ almost surely, so $B = 0$, $\mathrm{MSE}(g_c(\alpha))$ is a constant, and any $\alpha \in [0, 1]$ is optimal.

By the property of convex functions, the minimum value over a closed interval must be less than or equal to the value at the endpoints. Here, the endpoints correspond to the individual estimators: $\mathrm{MSE}(g_c(0)) = \mathrm{MSE}(g_e)$ and $\mathrm{MSE}(g_c(1)) = \mathrm{MSE}(g_t)$. Thus, it directly follows that:

$$\mathrm{MSE}(g_c(\alpha^\star)) \leq \min\{\mathrm{MSE}(g_t), \mathrm{MSE}(g_e)\}.$$

This completes the proof. $\qquad\square$

### A.3 PROOF OF THEOREM 3

*Proof.* The proof follows the derivation in Karimireddy et al. (2022, Appendix D.1), which applies the standard descent lemma and telescoping sum techniques (see e.g., Ghadimi & Lan, 2013; Ajalloeian & Stich, 2020).

**One-step descent.** Since $\mathcal{L}_m$ has an $L$-Lipschitz continuous gradient, for any $\theta, \theta' \in \mathbb{R}^d$ we have

$$\mathcal{L}_m(\theta') \leq \mathcal{L}_m(\theta) + \nabla_\theta \mathcal{L}_m(\theta)^\top (\theta' - \theta) + \frac{L}{2}\|\theta' - \theta\|^2. \tag{8}$$

Applying $\theta' = \theta_{k+1} = \theta_k - \eta g_c(\theta_k)$ to Equation (8), we get:

$$\mathcal{L}_m(\theta_{k+1}) \leq \mathcal{L}_m(\theta_k) - \eta \nabla_\theta \mathcal{L}_m(\theta_k)^\top g_c(\theta_k) + \frac{L\eta^2}{2}\|g_c(\theta_k)\|^2. \tag{9}$$

**Take conditional expectation.** Define $\mathbb{E}_k[\cdot] = \mathbb{E}[\cdot \mid \theta_k]$. Taking expectation of Equation (9) conditioning on $\theta_k$ gives

$$\mathbb{E}_k[\mathcal{L}_m(\theta_{k+1})] \leq \mathcal{L}_m(\theta_k) - \eta \nabla_\theta \mathcal{L}_m(\theta_k)^\top \mathbb{E}_k[g_c(\theta_k)] + \frac{L\eta^2}{2}\mathbb{E}_k[\|g_c(\theta_k)\|^2]. \tag{10}$$

**Bias-variance decomposition.** Let $a := \nabla_\theta \mathcal{L}_m(\theta_k)$, $b := \mathbb{E}_k[g_c(\theta_k)]$, and $\mathrm{Bias}_k := b - a$. The inner product is

$$a^\top b = \frac{1}{2}(\|a\|^2 + \|b\|^2 - \|\mathrm{Bias}_k\|^2). \tag{11}$$

Next, decompose $g_c(\theta_k)$ around its conditional mean:

$$g_c(\theta_k) = b + \underbrace{g_c(\theta_k) - b}_{D}, \quad \mathbb{E}_k[D] = 0. \tag{12}$$

Then since $\mathbb{E}_k[D] = 0$, the squared norm in Equation (10) satisfies

$$\mathbb{E}_k[\|g_c(\theta_k)\|^2] = \mathbb{E}_k[\|b + D\|^2] = \|b\|^2 + 2b^\top \mathbb{E}_k[D] + \mathbb{E}_k[\|D\|^2] = \|b\|^2 + \mathrm{Var}_k(g_c), \tag{13}$$

where $\mathrm{Var}_k(g_c)$ is the conditional scalar variance of $g_c$.

**Substitute and simplify.** By substituting Equations (11) to (13) into Equation (10), we have:

$$\mathbb{E}_k[\mathcal{L}_m(\theta_{k+1})] \le \mathcal{L}_m(\theta_k) - \frac{\eta}{2}\big(\|a\|^2 + \|b\|^2 - \|\mathrm{Bias}_k\|^2\big) + \frac{L\eta^2}{2}(\|b\|^2 + \mathrm{Var}_k(g_c))$$

$$= \mathcal{L}_m(\theta_k) - \frac{\eta}{2}\|a\|^2 - \frac{\eta}{2}(1 - \eta L)\|b\|^2 + \frac{\eta}{2}\|\mathrm{Bias}_k\|^2 + \frac{L\eta^2}{2}\mathrm{Var}_k(g_c).$$

Since $\eta \le 1/L$, we may drop the nonpositive term involving $(1 - \eta L)$ in the above equation to get a simpler upper bound:

$$\mathbb{E}_k[\mathcal{L}_m(\theta_{k+1})] \le \mathcal{L}_m(\theta_k) - \frac{\eta}{2}\|a\|^2 + \frac{\eta}{2}\|\mathrm{Bias}_k\|^2 + \frac{L\eta^2}{2}\mathrm{Var}_k(g_c). \tag{14}$$

**One-step gradient norm bound.** Rearranging Equation (14) yields:

$$\|\nabla_\theta \mathcal{L}_m(\theta_k)\|^2 \le \frac{2}{\eta}(\mathcal{L}_m(\theta_k) - \mathbb{E}_k[\mathcal{L}_m(\theta_{k+1})]) + \|\mathrm{Bias}_k\|^2 + \eta L\,\mathrm{Var}_k(g_c). \tag{15}$$

**Telescope over $K$ iterations.** Taking the total expectation of Equation (15), summing over $k = 0, \dots, K - 1$, dividing by $K$, and applying the law of total expectation gives:

$$\mathbb{E}\Big[\frac{1}{K}\sum_{k=0}^{K-1}\|\nabla_\theta \mathcal{L}_m(\theta_k)\|^2\Big] \le \frac{2}{K\eta}\sum_{k=0}^{K-1}(\mathbb{E}[\mathcal{L}_m(\theta_k)] - \mathbb{E}[\mathcal{L}_m(\theta_{k+1})])$$

$$+ \frac{1}{K}\sum_{k=0}^{K-1}\mathbb{E}[\|\mathrm{Bias}_k\|^2 + \eta L\,\mathrm{Var}_k(g_c)]. \tag{16}$$

Telescoping the first sum in the right-hand side of the above equation yields the exact bound in Equation (3).

**Apply uniform bounds.** Finally, applying the uniform bounds $\|\mathrm{Bias}_k\| \le B_c$ and $\mathrm{Var}_k(g_c) \le \sigma_c^2$ to Equation (16), and using $\mathbb{E}[\mathcal{L}_m(\theta_K)] \ge \mathcal{L}^*$, we arrive at the simplified bound of this theorem, completing the proof. $\qquad\square$

### A.4 Proof of Conditional Bias-Variance Decomposition in Equation (5)

**Proposition 1** (Conditional Bias-Variance Decomposition). *Let $g_c(\alpha, \theta_k) \in \mathbb{R}^d$ be a stochastic estimator of the true marginal gradient $\mu_k := \nabla_\theta \mathcal{L}_m(\theta_k)$ at iterate $\theta_k$. Then the conditional mean squared error (MSE) of $g_c(\alpha, \theta_k)$ can be decomposed into its squared conditional bias and conditional variance as:*

$$\mathrm{MSE}_k(g_c(\alpha, \theta_k)) = \mathbb{E}_k\big[\|g_c(\alpha, \theta_k) - \mu_k\|^2\big] = \big\|\mathrm{Bias}_k(g_c(\alpha))\big\|^2 + \mathrm{Var}_k(g_c(\alpha)),$$

*where*

$$\mathrm{Bias}_k(g_c(\alpha)) := \mathbb{E}_k[g_c(\alpha, \theta_k)] - \mu_k,$$

$$\mathrm{Var}_k(g_c(\alpha)) := \mathbb{E}_k\big[\|g_c(\alpha, \theta_k) - \mathbb{E}_k[g_c(\alpha, \theta_k)]\|^2\big].$$

*Proof.* Introduce the decomposition

$$g_c(\alpha, \theta_k) - \mu_k = \big(g_c(\alpha, \theta_k) - \mathbb{E}_k[g_c(\alpha, \theta_k)]\big) + \big(\mathbb{E}_k[g_c(\alpha, \theta_k)] - \mu_k\big) = D_k + \mathrm{Bias}_k(g_c(\alpha)),$$

where $D_k = g_c(\alpha, \theta_k) - \mathbb{E}_k[g_c(\alpha, \theta_k)]$. Expanding the squared norm and taking the conditional expectation yields

$$\mathbb{E}_k\big[\|g_c(\alpha, \theta_k) - \mu_k\|^2\big] = \mathbb{E}_k\big[\|D_k + \mathrm{Bias}_k(g_c(\alpha))\|^2\big]$$

$$= \mathbb{E}_k[\|D_k\|^2] + \|\mathrm{Bias}_k(g_c(\alpha))\|^2 + 2\,\mathbb{E}_k[D_k]^\top\mathrm{Bias}_k(g_c(\alpha)).$$

By definition, $\mathbb{E}_k[D_k] = 0$, so the cross-term vanishes. Moreover, $\mathbb{E}_k[\|D_k\|^2] = \mathrm{Var}_k(g_c(\alpha))$ by the definition of conditional variance. Combining these results gives

$$\mathbb{E}_k\big[\|g_c(\alpha, \theta_k) - \mu_k\|^2\big] = \|\mathrm{Bias}_k(g_c(\alpha))\|^2 + \mathrm{Var}_k(g_c(\alpha)),$$

which completes the proof. $\qquad\square$

Table 3: Variance ratios of log-probability and sequence-length comparing *Thinking* to *NoThinking*.

| Model | Variance ratio (log $p$) | Variance ratio (length) |
|---|---|---|
| R1-Qwen-7B | 10.17 | 2.91 |
| R1-Qwen-1.5B | 3.68 | 1.32 |
| R1-0528-Qwen3-8B | 1.23 | 1.11 |

Table 4: Mean length ratio and NLL comparing *Thinking* to *NoThinking*.

| Model | Mean length ratio | NLL$_{\text{Think}}$ | NLL$_{\text{No}}$ | $\Delta$NLL | % increase |
|---|---|---|---|---|---|
| R1-Qwen-7B | 3.17 | 0.384 | 0.316 | 0.068 | 21.5% |
| R1-Qwen-1.5B | 2.72 | 0.572 | 0.496 | 0.076 | 15.4% |
| R1-0528-Qwen3-8B | 1.86 | 0.456 | 0.381 | 0.075 | 19.8% |

### A.5 PROOF OF THEOREM 4

*Proof.* Under the condition $\eta L = 1$, the per-step convergence error simplifies to:

$$E_k(\alpha) = \|\text{Bias}_k(g_c(\alpha))\|^2 + \text{Var}_k(g_c(\alpha)),$$

which is precisely the conditional MSE in Equation (5). Since $E_k(\alpha) \equiv \text{MSE}_k(\alpha)$, the minimizer of one is necessarily the minimizer of the other. $\qquad\square$

## B ANALYSIS OF LRM'S LOG-PROBABILITY AND SEQUENCE-LENGTH STOCHASTICITY

In this section, we empirically quantify the stochasticity introduced by sampling reasoning traces, compared with trace sampling disabled. The results support the claim that stochastic trace sampling increases gradient variance, motivating our BVPO.

### B.1 SETUP

For each question, we sample five responses under two settings: reasoning-trace sampling enabled (*Thinking*) and disabled (*NoThinking*). Across the five samples, we compute the mean and variance of the joint log-probability and the sequence length, as well as the negative log-likelihood (NLL). We then average these per-question statistics over all questions.

### B.2 RESULTS

**Trace Sampling Increases Variance.** In Table 3, we report the variance ratios of log-probability and sequence-length comparing *Thinking* to *NoThinking*. We can see that reasoning trace generation increases dispersion in both joint log-probabilities and output lengths. Relative to *NoThinking*, *Thinking*'s variance of joint log-probabilities rises by up to 10.17 times while the variance of length rises by up to 2.91 times.

**Trace Sampling Increases Sequence Length and NLL.** Table 4 complements variance with length ratio and token-level predictability. *Thinking* yields substantially longer outputs by up to 3.17 times. Per-token NLL increases by up to 21.5%. Because NLL is normalized by length, this worsening cannot be attributed solely to longer sequences; tokens generated with reasoning trace generation enabled are intrinsically harder to predict. In the preference optimization context, noisier tokens and longer trajectories compound to amplify gradient variability, reinforcing the need for an estimator that explicitly manages the bias–variance trade-off.

**Within-*Thinking* Localization.** In Table 5, we additionally provide an analysis of stochasticity within the sampled *Thinking* responses. Within *Thinking*, the reasoning trace accounts for the majority of tokens (55–64% by length), and its per-token NLL is 1.15–1.73 times higher than the final

Table 5: Within-*Thinking* decomposition into trace vs. answer segments. NLL in nats/token.

| Model | Trace token share | $\text{NLL}_{\text{trace}}$ | $\text{NLL}_{\text{answer}}$ | NLL ratio (trace/answer) |
|---|---|---|---|---|
| R1-Qwen-7B | 0.639 | 0.453 | 0.261 | 1.73 |
| R1-Qwen-1.5B | 0.612 | 0.679 | 0.403 | 1.69 |
| R1-0528-Qwen3-8B | 0.549 | 0.484 | 0.422 | 1.15 |

answer segment. This shows that the trace segment is both larger and less predictable, so fluctuations in $\log p(r, y \mid x)$ are predominantly trace-driven. These observations align with BVPO's design choice to incorporate an empty-trace component: by construction it is agnostic to trace sampling, thereby reducing the conditional variance term that dominates in *Thinking* mode and tightening the convergence floor in Equation (4).

These statistics provide strong empirical evidence for the stochasticity caused by reasoning trace sampling, highlighting the instability of the standard trace-based gradient estimator $g_t$ and motivating the need for our BVPO.

## C   EXPERIMENT DETAILS

### C.1   HYPERPARAMETER SETTINGS

We use a consistent batch size of 128 and train all methods for 1 epoch in all settings. The AdamW optimizer (Loshchilov & Hutter, 2019) is used. The max sequence length is set to 4096 and a cosine learning rate schedule with 10% warm-up steps is used. $\alpha$ for BVPO is set as 0.5 in our experiment. The hyperparameters for each method are grid-searched and are shown in Table 6 for DPO, Table 7 for SimPO, and Table 8 for our BVPO correspondingly. The training is conducted using 8 GPUs.

Table 6: The hyperparameters of DPO for each training setting.

| Setting | $\beta$ | learning rate |
|---|---|---|
| **DeepSeek-R1-Distill-Qwen-7B** | 0.01 | 7e-7 |
| **DeepSeek-R1-Distill-Qwen-1.5B** | 0.01 | 7e-7 |
| **DeepSeek-R1-0528-Qwen3-8B** | 0.01 | 7e-7 |

Table 7: The hyperparameters of SimPO for each training setting.

| Setting | $\beta$ | $\gamma$ | learning rate |
|---|---|---|---|
| **DeepSeek-R1-Distill-Qwen-7B** | 2.5 | 1.0 | 7e-7 |
| **DeepSeek-R1-Distill-Qwen-1.5B** | 2.5 | 1.0 | 7e-7 |
| **DeepSeek-R1-0528-Qwen3-8B** | 2.5 | 1.0 | 7e-7 |

Table 8: The hyperparameters of BVPO for each training setting.

| Setting | $\beta$ | learning rate |
|---|---|---|
| **DeepSeek-R1-Distill-Qwen-7B** | 0.01 | 7e-7 |
| **DeepSeek-R1-Distill-Qwen-1.5B** | 0.01 | 7e-7 |
| **DeepSeek-R1-0528-Qwen3-8B** | 0.1 | 7e-7 |

### C.2   EVALUATION DETAILS

For alignment benchmarks (AlpacaEval 2 and Arena-Hard), we set the maximum generation length to 8192 tokens. Following DeepSeek-AI et al. (2025), evaluation uses only the final answer part of each response. GPT-4o-2024-11-20 is used as the judge model.

For math reasoning benchmarks, we increase the maximum generation length to 32768 tokens to accommodate problems requiring extended reasoning and to ensure a sufficiently large context window.

### C.3 DATA GENERATION

For the trace-based set $\mathcal{D}_t$, we use the experimented models' official chat template to sample responses, which allows free-form reasoning trace generation.

> **Template for sampling $\mathcal{D}_t$**
>
> <|begin_of_sentence|><|User|>{QUESTION}<|Assistant|><think>

For the empty-trace set $\mathcal{D}_e$, we explicitly disable reasoning trace sampling by additionally appending </think> at the beginning of the assistant turn, since the special token </think> denotes the end of reasoning trace generation. For the sampled responses, we prepend the special token </think> so that they remain consistent with the official chat template that generates reasoning traces.

> **Template for sampling $\mathcal{D}_e$**
>
> <|begin_of_sentence|><|User|>{QUESTION}<|Assistant|><think></think>

## D LLM USAGE DISCLOSURE

In preparing this manuscript, we employed a large language model (LLM) as a writing assistant. Its use was strictly limited to enhancing clarity, readability, and grammatical correctness. Concretely, the LLM was used for rephrasing sentences to improve flow, suggesting alternative phrasings for technical descriptions, and converting tables into LaTeX format. All core scientific ideas, theoretical derivations, experimental results, and conclusions were developed and written solely by the human authors. The authors carefully reviewed and edited all LLM-assisted outputs and bear full responsibility for the final content and its scientific accuracy.

