# OpenReview forum: "From Noisy Traces to Stable Gradients: Bias--Variance Optimized Preference Optimization for Aligning Large Reasoning Models"
_ICLR.cc/2026/Conference — Submitted to ICLR 2026_

### Official Review · Reviewer_udkN · 2025-10-20

**Soundness:** 3
**Presentation:** 3
**Contribution:** 2
**Rating:** 4
**Confidence:** 4

**Summary:**

This work concentrates on the task of aligning large reasoning models with human preferences via RLHF/DPO. The authors argue that the gradient variance of the reasoning traces impacts the performance of conventional alignment methods, and propose BVPO to jointly consider both bias and variance. Specifically, an empty-trace gradient is computed and linearly merged with the original trace-based gradient, with MSE minimization to decide the optimal combination weights. In experiments, the proposed BVPO achieves better performance on both alignment and reasoning tasks.

**Strengths:**

1)	The proposed idea is simple and sound. This work is easy to follow.
2)	The authors have conducted various of downstream tasks to evaluate the methods’ performance on both alignment and reasoning tasks.
3)	The theoretical analysis part provides a good discussion on the MSE-based selections.

**Weaknesses:**

1)	The straightforward linear combination of L_t and L_e outperforms SimPO and DPO consistently on both alignment and reasoning tasks, which is impressive. Is the reason for the failure of baselines really due to the high variance caused by reasoning traces? Is it possible that the improvement of BVPO is mainly brought by extra prompts and additional training signals of g_e? Why does the effectiveness continue to improve in reasoning tasks?
2)	The authors should provide the exact value of \alpha in different settings. Moreover, the effectiveness of the MSE minimization should be verified (e.g., by providing experiments with different hyperparameter \alpha on both tasks).
3)	The authors conduct the experiments merely on Qwen series. The generalization ability of BVPO on other LLM structures should be evaluated.

**Questions:**

1)	Is there any better selection besides the empty trace that could further enhance the performance of BVPO? For instance, a “shorter reasoning prompt”?
2)	How to extend BVPO to RLHF besides DPO?

---

### Official Review · Reviewer_NDzH · 2025-10-29

**Soundness:** 2
**Presentation:** 3
**Contribution:** 2
**Rating:** 2
**Confidence:** 4

**Summary:**

This paper addresses aligning large reasoning models with human preferences, focusing on the instability caused by noisy reasoning traces. It introduces Bias–Variance Optimized Preference Optimization, combining high-variance trace-based gradients with low-variance empty-trace gradients to optimize the bias-variance trade-off. Theoretical results prove variance reduction and MSE-optimal mixing, while experiments on DeepSeek-R1 models show up to 7.8 points improvement on AlpacaEval 2 and 6.8 points on Arena-Hard, with a 4.0-point boost in math reasoning.

**Strengths:**

1. Offers rigorous proofs of variance reduction and MSE optimization, providing a solid foundation for BVPO's design.
2. Demonstrates significant alignment and reasoning improvements across diverse benchmarks, validating practical impact.
3. As a drop-in method, BVPO integrates easily with existing DPO pipelines, enhancing accessibility.

**Weaknesses:**

- The BVPO method is more like a data augmentation method than a improvement in algorithm. The theoretical analysis induced optimal value of $\alpha$, but it is never used in practical, and the authors simply chose 0.5
- Insufficient experiments. This work only compares with two baselines SimPO and DPO on ArenaHard and AlpacaEval 2.0. Some important baselines are missing, such as mixing thinking/non-thinking preference data
- Lack of analysis. There are limited analysis conducted, only evaluation on Math reasoning benchmarks. Some important ablation studies, such as the value of $\alpha$, $\beta$, were not included.

**Questions:**

- On the implementation of DPO and SimPO, did you augment the empty-trace datasets or only use it for BVPO? If so, there would be an unfair comparison issue since BVPO used twice more training data than the baselines.
- The 4096 max length is quite short for reasoning models. How do you deal with longer generations? Why the evaluation used 8192 max tokens which is not aligned with training?

---

### Official Review · Reviewer_SRsw · 2025-10-30

**Soundness:** 3
**Presentation:** 2
**Contribution:** 2
**Rating:** 4
**Confidence:** 3

**Summary:**

This paper targets aligning LRMs with human-preferred final answers while marginalizing intractable reasoning traces. It spotlights gradient variance caused by sampling a single trace and proposes BVPO, which optimally mixes high-variance trace gradients $g_t$ and low-variance empty-trace gradients $g_e$ by minimizing MSE to the true marginal gradient.  Theoretical analysis shows that BVPO reduces variance and improves convergence bounds. Empirically, BVPO outperforms strong baselines like DPO and SimPO on AlpacaEval 2 and Arena-Hard, and improves reasoning performance on math benchmarks.

**Strengths:**

1. The paper introduces a new method for preference optimization in LRMs, and derives a closed-form optimal mixing coefficient for gradient estimation.
2. The work is presented clearly: problem formulation, method derivation, and experiments are easy to follow.
3. The paper shows that simple gradient re-weighting simultaneously improves both chat quality and mathematical reasoning without extra data.

**Weaknesses:**

1. **Distribution shift of empty traces**
   Answers generated under the `<think></think>` condition come from a different prompt distribution than regular reasoning traces. The paper simply interpolates the two log-probabilities without measuring the bias induced by this shift, nor does it report accuracy or consistency of the empty-trace replies on a validation set.

2. **The optimal $\alpha^*$ relies on unknown quantities**
   The closed-form optimal weight $\alpha^*$ needs $Σ_{te}, b_t, b_e$, etc. In practice these expectations are replaced by mini-batch estimates whose error is not analyzed. If the estimator variance is large, the “optimal” weight can deviate far from the theoretical value and even inject extra noise; the manuscript provides no bound or ablation on this estimation error.

3. **Experiments limited to SFT-only checkpoints**
   All results are obtained from DeepSeek-R1-Distill models that only undergo supervised fine-tuning. Their traces are short and well-formatted, so gradient variance is relatively mild. After large-scale RL (especially RLVR), reasoning models produce much longer and more exploratory chains, amplifying noise. Whether BVPO still suppresses variance under this harder regime remains untested.

**Questions:**

1. **Empty-trace distribution shift**

    Have you measured the KL divergence or any task-accuracy drop between regular and empty-trace conditions?

2. **RL-checkpoints test**

    Can you report the result on a fully RL-tuned model (e.g. OpenMath-Nemotron-1.5B) to show whether BVPO still works when traces are longer and noisier?

---

### Official Review · Reviewer_xshe · 2025-10-31

**Soundness:** 3
**Presentation:** 3
**Contribution:** 3
**Rating:** 6
**Confidence:** 2

**Summary:**

This paper addresses the challenge of aligning Large Reasoning Models (LRMs) with human preferences. The authors identify that the standard approach of using trace-based gradients suffers from high variance due to stochastic reasoning trace sampling. They propose BVPO (Bias-Variance Optimized Preference Optimization), which combines a high-variance trace-based gradient estimator with a low-variance empty-trace estimator through convex combination. The method provides theoretical guarantees on variance reduction and MSE optimality, and demonstrates empirical improvements of up to 7.8 points on AlpacaEval 2 and 6.8 points on Arena-Hard, while also improving math reasoning performance by up to 4.0 points across six benchmarks.

**Strengths:**

1. **Problem Identification**: The paper makes a valuable contribution by clearly identifying and quantifying gradient variance as a key bottleneck in LRM alignment. The empirical analysis in Appendix B provides strong evidence with variance ratios and NLL decomposition.

2. **Theoretical Rigor**: The theoretical framework is comprehensive, covering variance reduction, MSE optimality, and convergence guarantees. The proofs appear sound and build appropriately on established optimization theory.

3. **Empirical Validation**: Experiments are thorough, testing on three model sizes (1.5B, 7B, 8B parameters) across both alignment benchmarks (AlpacaEval 2, Arena-Hard) and reasoning benchmarks (AIME, AMC, Minerva, OlympiadBench, MATH-500).

4. **Practical Impact**: The consistent improvements over strong baselines and the surprising benefit to math reasoning suggest real practical value.

**Weaknesses:**

**Dataset Cost and Efficiency**:
   - Requires generating two separate preference datasets (D_t and D_e)
   - No analysis of whether the same improvement could be achieved with the same total data budget using only one approach
   - No discussion of computational overhead during training

 **Limited Scope**:
   - Only tested on DeepSeek-R1 family models
   - All models share the same base architecture (Qwen)
   - No exploration of other LRM architectures or reasoning approaches

**Questions:**

see Weaknesses

---

### Meta-Review · Area_Chair_uvtV · 2025-12-17

**Summary:**

This paper proposes Bias–Variance Optimized Preference Optimization (BVPO), aiming to address the instability caused by noisy inference trajectories in large-scale inference models. It optimizes the trade-off between bias and variance by combining high-variance trajectory gradients and low-variance empty trajectory gradients. The paper theoretically proves variance reduction and MSE-optimal mixture. However, the reviewers pointed out that BVPO is more like a data augmentation method rather than an algorithm improvement.  The experiments and analysis are insufficient. Additionally, there is a potential unfair comparison issue in implementation and data quantity.

**Reviewer Concerns:**

Authors do not reply the problems raised by reviewers.

**Reviewer Scores:**

The scores will remain unchanged.

---

### Decision · Program_Chairs · 2026-01-26

Reject